# Ultrasound for Gaze Estimation—A Modeling and Empirical Study [note 1]

**DOI:** 10.3390/s21134502

**Published:** 2021-06-30

**Authors:** Andre Golard, Sachin S. Talathi

**Affiliations:** Facebook Reality Labs, Redmond, WA 98052, USA; agolard@fb.com

**Keywords:** eye tracking, gaze estimation, ultrasound, CMUT, machine learning, Gradient Boosted Regression Trees, Comsol Modeling

## Abstract

Most eye tracking methods are light-based. As such, they can suffer from ambient light changes when used outdoors, especially for use cases where eye trackers are embedded in Augmented Reality glasses. It has been recently suggested that ultrasound could provide a low power, fast, light-insensitive alternative to camera-based sensors for eye tracking. Here, we report on our work on modeling ultrasound sensor integration into a glasses form factor AR device to evaluate the feasibility of estimating eye-gaze in various configurations. Next, we designed a benchtop experimental setup to collect empirical data on time of flight and amplitude signals for reflected ultrasound waves for a range of gaze angles of a model eye. We used this data as input for a low-complexity gradient-boosted tree machine learning regression model and demonstrate that we can effectively estimate gaze (gaze RMSE error of 0.965 ± 0.178 degrees with an adjusted $R^{2}$ score of 90.2 ± 4.6).

## 1. Introduction

Most current eye tracking methodologies use video to capture the position of the iris and/or reflected lights sources–glints [1]. As such, these methods can be affected by ambient light [2], which is particularly true for use cases such as augmented reality with eye glasses. Other light-based methods such as scanning lasers [3], third Purkinje images [4] and directional light sensors [5] can likewise be affected. Speed can also be limited, especially in wearables, where operating a camera at high speed (on the order of 100 Hz or above) would imply high power consumption. At these speeds the camera-based sensors can capture fixations but not other parameters such as saccades, which have been implicated as a markers of schizophrenia spectrum in at-risk mental states [6] as well as other neurological disorders [7]. Fast eye tracking is required for measuring saccades. Current devices capable of measuring saccades are designed for laboratory use, and tend to lack portability [8]. The possibility of using ultrasound for eye tracking has been raised in a patent [9] and there exist studies that use eye-tracking to assist ultrasound procedures [10]. However, to the best of our knowledge, there is no report on experimental study to empirically demonstrate the feasibility for gaze estimation using ultrasound sensors.

A recent paper explored the possibility of using non-contact ultrasound sensors to track fast eye movements [11]. The work focused on the development of a finite element simulation model to investigate the use for ultrasound time of flight data to track fast eye motions. The simulation model is based on a setup made of four transducers positioned perpendicular to the cornea. Distances are measured with each transducer based on the time for it to receive the reflection of its own signal. Given the cornea protrudes, this time changes with the gaze angle. For implementing this simulation setup in any form of glasses-form factor device, the device needs to be precisely positioned relative to the eye. However, we are interested in applications for eye tracking in augmented reality (AR) and virtual reality (VR), where user-specific placement of the sensors is not possible (in AR/VR the eye tracking system will be fixed and the position of the eye will vary from user to user, which means alignment will vary).

It is also to be noted that the modeling in [11] was done in the absence of occlusions (such as eyelids). Eye occlusions are known to be problematic for eye tracking systems in general [12]. Furthermore, the authors [11] chose to model standard 40 kHz transducers. While these would be advantageous in terms of minimizing attenuation in air, such a system may be subject to interference from range-finding applications (typically in the 40–70 kHz range). Common range finding systems lack the resolution and short distance sensing capabilities required for eye tracking (the typical sensing range would be in meters with a resolution of 1 cm). Another concern for our application of interest is size. Devices would need to fit in glasses frames. Capacitive Micromachined Ultrasonic Transducers (CMUTs) operating at 500 kHz–2 MHz [13] provide the range, resolution and size that is suitable for use in VR and AR devices. This type of transducer has found numerous medical applications in both imaging and therapy [13], which are applications for contact ultrasound.

Here, we use the CMUTs for remote sensing as airborne transmitters and receivers. In this mode, the difference in impedance between air and tissue means over 99 percent of the ultrasound signal will be reflected by the eye surface [11]. As such, the size of transducers was a primary concern for our choice of CMUTs for the proposed study and concerns related to test bench size and power consumption did not drive our investigations.

In order to systematically investigate the feasibility of near-field ultrasound sensing for eye-tracking with an AR form-factor device, we first did our own finite-element-modeling study using acoustic rays for 1.7 MHz transducers configured on AR glasses. We compared directional and omnidirectional transmit and receive configuration for the sensors to determine where we would expect to see a meaningful signal around glasses frames for a source placed near the glasses branch. We then built a series of table top test bench systems to (a) verify our ability to accurately measure distances in the appropriate range, (b) characterize the transducers, and (c) generate data to be used in a machine learning model to estimate gaze. As such we focus on empirically testing the hypothesis that ultrasound sensors can be used for gaze estimation in the presence of occlusions. We note that in the context of our experiments, gaze is defined by the static orientation of model eye on the goniometer. We demonstrate that ultrasound time of flight and amplitude signals can be leveraged to track gaze in such conditions. In particular, we train a regression model using gradient boosted decision trees to estimate the gaze vector given the set of ultrasound time-of-flight and amplitude signals captured by the CMUT receivers. The nonlinearities introduced by occlusion artifacts make the task of regressing gaze directly from recorded signals non-trivial and we believe that a nonlinear regression model trained on the collected data is best suited to extract the relvant signals for gaze estimation. Results show that the trained model produces a regression $R^{2}$ score of 90.2 ± 4.6% and a gaze RMSE error of 0.965 ± 0.178 degrees.

## 2. Materials and Methods

In this section, we describe the set up for acoustic ray tracing modeling, the benchtop experimental setup for data collection, the signal processing steps to extract the ultrasound time of flight and amplitude signals, and the machine learning framework adopted to train a gaze estimation model.

### 2.1. Modeling

Ultrasound is modeled as rays released all at once from a single point. Their position is updated at fixed time intervals. We did this so we could trace the path of signals reaching the receiver and determine if they were reflections from the cornea or the skin or glasses. We used the acoustic ray tracing features of COMSOL Multiphysics software (https://www.comsol.com/release/5.5). We used a fixed value of 343 m/s for the speed of sound in air. We accounted for an acoustic wave attenuation in air that corresponds to 1.7 MHz. The absorption attenuation coefficient is 470 (dB/m). To estimate the reflection of the signal we used the formula $R={\left(\frac{Z_{2}-Z_{1}}{Z_{2}+Z_{1}}\right)}^{2}$ where R is a measure for the fraction of sound reflected and Z${}_{1}$ and Z${}_{2}$ are the impedance of the two media [11]. The acoustic impedance of a medium are its density times the speed of sound. We used a density of 1 kg/m${}^{3}$ for air. The densities of the solids range from 911 kg/m${}^{3}$ for the tear film to 2580 kg/m${}^{3}$ for glass, with 1051 kg/m${}^{3}$ for the cornea. The speed of sound in the solids ranges from 1450 m/s in fat to 4500 m/s in glass. Based on these values we estimate 99.87–99.99 percent of the signal will be reflected. These calculations guided our decision to assume 100% reflection of ultrasound waves of eye in our modeling.

We used a scanned eye surface obtained with an Eye3D scanner (Transfolio, Marina del Rey, CA, USA). A fit of cornea with a sphere shows a radius of 5.65 mm. The surface was smoothed, and the mesh size adjusted using Autodesk Meshmixer (https://www.meshmixer.com). We used it to create gaze variants: straight, ±20 degrees in the vertical direction, ±30 degrees horizontal.

We used a scanned face and glasses designed in Solidworks to create the eye box (the space in which rays will propagate). Locations for the transducers are shown in Figure 1. These positions were arbitrary. (The Comsol model was built by Veryst, Needham Heights, MA, USA).

### 2.2. Benchtop Setup

We designed a series of three test benches to evaluate distance measurements, signal attenuation, transducer directionality, and our ability to estimate gaze.

In terms of electronics and data acquisition, all test benches are based on a CMUT evaluation kit from Fraunhofer IPMS (Dresden, Germany). This test kit is comprised of CMUT transducers (1.74 MHz), an amplifier, bias-tee, and associated software. These transducers fit our size and power requirements.

We first verified our ability to measure distances, as well as the signal decay due to attenuation in air given that ultrasound signal attenuation is significant at MHz frequencies [14]. We used a setup consisting of a pair of transducers aimed at a flat target attached to a linear translation stage (Test bench 1, Figure 2A).

Next we tested the emission properties of the transducers. Our CMUTs are comprised of an array of cells connected to a single electrode and a single counter electrode. As such they act as a fixed phased array, which is expected to exhibit directionality. We tested this using a fixed transducer and one on a rotating stage (Test bench 2, Figure 2B). The Tx transducer was rotated in 1 degree increments and the amplitude of the Rx signal was recorded.

Our third test bench is designed for gaze estimations (Figure 3A). As noted earlier, we define gaze in terms of the static orientation of model eye on the goniometer. The transducer side is on the right. We used a pair of transducers (one in transmit mode and one receiver) mounted on rotating stages to allow us to mimic multiple locations around a ring (or glasses frame). We acquired data for all transmit and receive locations covering 360 degrees in 10 degree increments (Figure 3C).

On the target side (left part of Figure 3A), a standard sphere on sphere model eye (cornea radius 7.8 mm, sclera radius 11.925 mm, offset 5.6 mm) was mounted on a goniometer (Thor Labs). Note these dimensions differ slightly from the scanned eye used for modeling. This does not affect our findings, see discussion. Gaze angles were set in one degree increments between ±5 degrees in both up/down (ϕ) and left/right (θ) directions.

Occlusions (known to affect eye trackers) were added for realism. This is a step forward from previous modeling which totally ignored occlusions. We did not model or attempt to integrate eyelashes. Our occlusions consisted of a partial scanned face printed in flexible material (A40 durometer Polyjet) with a cavity to accommodate the model eye (Figure 3B). This was mounted in front of and against the model eye and allowed the eye to move freely.

Our test signal consisted of a train of seven oscillations at 1.74 MHz, repeated at 2 kHz. The transmitter was moved to positions around a 180 degree arc opposite the receiver (−90, −80, ..., 80, 90), Figure 3C. Fifty runs were recorded for each transducer position. The series was repeated for all static goniometer positions. The received signal was digitized at 80 MHz.

### 2.3. Data Analysis

#### 2.3.1. Feature Engineering

The raw signal carries too much noise to allow for accurate peak time and amplitude measurements. Improvements are possible (data not shown). For this proof of concept we used averaging and filtering. In Figure 4A, we show one raw trace, $x_{i}^{r}(t,\theta =0,$
ϕ=0) ($i\in \left[0,49\right]$ and r∈[−90,−80,⋯80,90]), for the ultrasound signal captured at the receiver, in response to a single test signal emitted by the transmit CMUT transducer. Figure 4B, shows the average of ten traces, defined as ${\bar{x}}_{k}^{r}\left(t\right)=0.1\sum_{j}^{j+10}x_{j}\left(t\right)$ ($k\in \left[0,4\right]$). The ultrasound time of flight, $\tau_{k}^{r}\left(\theta ,\phi \right)$, and amplitude, $a_{k}^{r}\left(\theta ,\phi \right)$, signal is estimated for each ${\bar{x}}_{k}^{r}\left(t,\theta ,\phi \right)$ as follows: the signal, ${\bar{x}}_{k}^{r}\left(t,\theta ,\phi \right)$ is band-pass filtered in the frequency range, [1.6 MHz–1.9 MHz] using a Butterworth filter of order 4 to generate the filtered version, $f\left({\bar{x}}_{k}^{r}\right)\left(t,\theta ,\phi \right)$. In Figure 4C, we show the trace for $f^{2}\left({\bar{x}}_{k}^{r}\right)\left(t,\theta =0,\phi =0\right)$. The ultrasound time to peak $\tau_{k}^{r}\left(\theta ,\phi \right)$ and the amplitude, $a_{k}^{r}\left(\theta ,\phi \right)$ is obtained by considering a time window of 45 μsaround the time instance of peak value for $f^{2}\left({\bar{x}}_{k}^{r}\left(t,\theta ,\phi \right)\right)$ and finding the first instance of the peak value for ${\bar{x}}_{k}^{r}\left(t,\theta ,\phi \right)$ within the considered time window. The detected peak value represents the amplitude signal $a_{k}^{r}\left(\theta ,\phi \right)$ and the time to peak, $\tau_{k}^{r}\left(\theta ,\phi \right)$. Thus, for each position Y=(θ,ϕ) of the model eye on the goniometer, we obtain a set k = 5 feature vectors $X\in R^{36}={\left\{a^{r},\tau^{r}\right\}}_{r=[-90,-80\cdots 80,90]}$ per experimental run. In order to collect sufficiently robust dataset and also to account for changes in day to day environmental fluctuations, we conducted a total of 9 experiments spanning a period of 9 days. In total, for each position, Y, on the goniometer, we were able to compile a set of 9×5 feature vectors, X, and our goal for ultrasound based eye tracking is to learn a regression model, H:X→Y; that is, given the ultrasound sensor time of flight and amplitude data, estimate two-dimensional eye gaze coordinates.

In Figure 5A,B, we plot the distribution of $\tau^{r}\left(0,0\right)$ and $a^{r}\left(0,0\right)$ respectively. In the last sub-plot for each of the figures we show how the mean time-of-flight and the mean amplitude signal changes as function of the position of the receiver transducer. It is worth noting that while the time-of-flight signal falls of symmetrically from the center of gaze, the amplitude signal peaks at receiver r=20, a result of occlusion from the nose-pad. We also note of the distribution spread for the time-of-flight and the amplitude signal captured by each receiver, which may be the result of measurement noise with our test-bench.

#### 2.3.2. Gradient Boosted Regression Trees

From a machine learning perspective, the task of learning a gaze estimation model *H* is categorized as a supervised regression problem. Gradient Boosting Regression Trees (GBRT) are a powerful class of boosting algorithms for classification and regression tasks, which combine output from several weak learners into a powerful estimator. Specifically, GBRT considers additive models of the form: $F_{m}\left(x\right)=F_{m-1}\left(x\right)+h_{m}\left(x\right)$, where $h_{m}$ are the basis functions modeled as small regression trees of fixed size. For each boosting iteration, a new boosting tree is added to the GBRT model, *F*. For our problem, we train two separate GBRT models to independently estimate the response in the horizontal and vertical dimensions: Y=(θ,ϕ) as function of the input features, $X=(\tau^{r},a^{r})$. Assuming the GBRT model is comprised of *M* regression trees with $T_{m}$ leaf nodes per regression tree, the GBRT model for each of the gaze regressor is given as: $F^{y}\left(X,w^{y}\right)=w_{0}^{y}+\sum_{m=1}^{M}\sum_{j=1}^{T_{m}}w_{jm}^{y}I\left(X\in R_{jm}^{y}\right)$, where y={θ,ϕ} and $R_{jm}^{y}$ represents the *j*th disjoint partitioning of the input space for the mth regression tree for the regressor variable, *y*. The GBRT model weights are estimated from data as follows: $w^{*}$= ${}_{w}\frac{1}{N}\sum_{i}^{N}L\left(y_{i},F\left(X_{i},w\right)\right)$ where, *L* is the squared error loss function. For an exhaustive description of GBRT, see [15,16].

Our choice of GBRT as a choice of regression model to predict gaze is motivated by the fact that occlusions introduce non-linearities for localizing gaze given a set of ultrasound time-of-flight and amplitude signals. In order to evaluate the utility for using a nonlinear regression model, we also train a linear regressor for gaze estimation (see Table 1).

Both the GBRT and linear regression models are trained to minimize the mean-squared-error between the estimated gaze-vector and the predicted gaze-vector and we report model performance in terms of root-mean-squared model error on a 5-fold cross-validation set. In addition we report the adjusted-$R^{2}$ as a goodness-of-fit measure for regression models.

## 3. Results

In this section we present findings from our modeling study as well as experiments conducted using the three benchtop setups described in Section 2.2.

We begin by presenting our findings on the CMUT sensor characterization. Data collected using test bench setup 1 allowed us to investigate the decay characteristics of the ultrasound signal in air, see Figure 6A. As expected, the ultrasound signal decays exponentially as a function of distance. An extrapolated fit shows it decays to zero. The distance axis shows the distance between the pair of transducers and the target (Figure 2A). Actual travel distance is twice this measurement. The range is similar to the distances for transducers mounted on eye glasses frames, our use case scenario.

Data collected using test bench 2 (Figure 2B) allowed us investigate whether the CMUT transducers exhibit directionality. Our findings are reported in Figure 6B. The CMUT transducers indeed exhibit directionality with an emission cone of 10 degrees. This applies to the transducers in both transmit and receive mode.

Based on the above findings we conclude that the strength of ultrasound signal at the receiver CMUT transducer will depend on two factors: distance and incident angle. As such, we believe that the amplitude of the ultrasound signal at the receiver contains relevant information to contribute to our ability to estimate gaze and as shown below, our findings indeed support this claim.

Our modeling study explored two situations: an omnidirectional transducer and one that mimics the properties of our CMUTs, see Figure 7. 131,072 rays are released from a point source in each case. The rationale for exploring the two situations is that while our CMUTs fit our needs, single crystal piezo transducers may provide a robust, inexpensive alternative. They are omnidirectional but can be turned into a directional device by adding baffles. In terms of size, they would be slightly larger (2.5 mm instead of 1 mm in our frequency range).

We implemented the sensor native curve by releasing rays with a uniform density distribution and assigning weight functions to the rays based on rays angle of emission and reception. The weight function of the rays is cos(min(alpha*(90/15),90°). Alpha is the angle between the ray and transducer direction.

In the directional case we used a similar approach to account for the receiver native curve. We assigned a similar weight function to the acoustic rays that reach receivers based on the angle between the incoming acoustic rays and the sensor direction of each receiver. Therefore, each ray has two weight functions. One weight function is assigned initially when the ray is released, another weight function is assigned when the ray is detected by a receiver. The product of the two weight functions is applied. If the angle between a ray (that reaches a sensor) and the receiver direction is more than 15 degrees then the ray is not detected (its weight function is zero). If this angle is zero then the weight function is 1.

Figure 8 shows a comparison of the predicted signal at our sensor locations for directional and omnidirectional transducers. The left and right panels correspond to thirty degree rotations of the eye to the left (towards the nose) or right. In the case of omnidirectional transducers the differences between gazes are small. Differences are more pronounced for directional transducers. Peaks are also better defined with directional transducers. Late peaks resulting from longer paths due to multiple reflections are minimized. It is to be noted that such late peaks would be ignored in our analysis, as we only use the time to peak and peak amplitude for the first peak detected in a given channel. With the same total number of rays (transducer power), receiver sensors with a directional transducer have higher signal strength than receiver sensors with an omnidirectional transducer. We ran the same models for a straight gaze as well as up/down twenty degree rotations (data not shown), and obtained similar results. Taken together the directional transducers perform better to resolve gaze.

Next we looked at where on the frame we might detect a signal, and why. Figure 9, left, shows signal intensity around the frame. Areas in red have a higher chance of detecting rays reflected from the eye. Rays reflected off the glasses or skin are ignored. The center panel shows the path taken by the rays that reach receiver 6. Some of the rays arrive after multiple reflections from the skin and glasses. The right panel provides a detailed view of the direction of rays reaching receiver 6 (sphere). Sensor direction is shown with the solid black line. The majority of these rays will not be detected by the receiver due to the narrow angle of detection dictated by the receiver native curve. If we were using omni-directional transducers, rays arriving after two or more reflections would broaden the signal or create multiple peaks. Directional transducers allow us to reject unwanted signals before they are counted.

We next report findings from training a GBRT model on data collected using the third test bench setup (see Figure 3). For each model eye position on the goniometer, θ,ϕ, for a fixed receiver transducer position (180 degrees) and for a set of 19 transmit transducer positions, we fire the ultrasound test signal 50 times, at 2 kHz and record the raw receiver signal (see Figure 4 top row). In order to increase the strength of ultrasound response at the receiver we average 10 traces of the raw response signals at a time, to effectively generate 5 averaged ultrasound response signals, in effect acquiring data at 200 Hz. The averaged response signal is passed through a Butterworth bandpass filter and we extract two ultrasound signal features: time of flight (τ) and the amplitude at peak (*a*), as explained in Section 2.3. In total for each model eye position, we generate a total of 45 samples for each model eye position on the goniometer over the duration of the study. For the set of 36 model eye positions, we produce a total of 1620 data samples.

We train a GBRT model on these data samples, performing a 5-fold cross-validation study. The model performance is reported using an adjusted R${}^{2}$ score [17] and the gaze RMSE error in degrees. Hyper-parameter search on the GBRT model parameters that produced the best adjusted R${}^{2}$ score for 5-fold cross-validation are reported in Table 2. We obtain gaze RMSE error of 0.965 ± 0.178 and mean adjusted R${}^{2}$ score of 90.2% with a standard deviation of 4.6, suggesting that almost 90% of the data fit the regression model. We also perform a similar analysis using a linear regression model and the results are reported in Table 1. Nonlinear modeling of the problem through GBRT produced an improvement in performance for RMSE of ≈ 18% and goodness-of-fit improvement of ≈5.7%, in support of our claim that the occlusions introduce nonlinearities in the ultrasound signals captured by the CMUT receivers, that can be best captured using a nonlinear regression model.

Residuals analysis confirmed that the estimates obtained using the GBRT model are un-biased (data not shown). In Figure 10, we show the plot of the fraction of GBRT estimated gaze values that fall within an epsilon-ball of given radius (degrees). We see that ≈50% of estimated gaze values fall within an epsilon ball of radius 0.8 degrees and ≈90% of estimated gaze values fall within an epsilon-ball of radius 2 degrees. Based on these findings, we conclude that using CMUT ultrasound sensors, we can expect gaze resolution of up to 2 degrees.

In Figure 11A,B, we show feature importance for the GBRT tree models trained to estimate the model eye gaze coordinates, θ (horizontal gaze) and ϕ (vertical gaze). We can see that the top two features for both horizontal and vertical gaze GBRT model are time of flight ultrasound signal. It has been our observation that while the time of flight component of ultrasound signal contains dominant information signal to estimate gaze (95% contribution to the regression score), the amplitude signal is also an important contributor for GBRT model to produce an adjusted-R${}^{2}$ score close to 90%. In order to test this observation, we trained GBRT model using just the ultrasound time-of-flight feature and another GBRT model using just the ultrasound amplitude feature. The findings are: GBRT model trained using time-of-flight features, produces an adjusted R${}^{2}$ score of 85.4 ± 5.2, where as the GBRT model trained using only the amplitude feature produces an adjusted R${}^{2}$ score of 78.6 ± 8.2. In Figure 9C, we show the mean-RMSE error (across all CV-folds) for the GBRT model. The error is biased towards the lower half of vertical gaze, primarily resulting from occlusions.

## 4. Discussion

This study is the first experimental demonstration of use for ultrasound sensors in gaze estimation. We show that ultrasonic transducers can effectively produce signals useful to resolve eye gaze, as defined by the static orientation of model eye on a goniometer, within the range tested, ±5 degrees in both up/down (θ) and left/right (ϕ) directions. This range reflects the full deflection of our goniometer. We plan on expanding the range in future studies.

Prior to embarking on our experiments with bench-top setup we conducted ray tracing modeling. This modeling helped us refine our test bench design, procedures, and analysis. First, it pointed to the utility of directional over omnidirectional transducers. Second, it informed us on where, given a source location, we can expect a signal around the glasses frame. Finally, it provided information on the signals we need to measure from our bench-top experiments: the time to peak (indicative of distance traveled), and the amplitude. Due to attenuation in air the amplitude decreases with travel distance. Our modeling indicated that amplitude also carries a signal based on the angle of incidence. This is further evidence for using directional instead of omnidirectional transducers.

Our GBRTs show that both amplitude and time of flight contribute to our ability to estimate gaze. This is a new finding as previous modeling work dealt with time of flight alone. As mentioned in our modeling section, two factors contribute to amplitude: attenuation and the incident angle of the incoming sound. One way to compensate for attenuation is to use the time-gain correction built in our amplifier, increasing gain over time to compensate for the signal attenuation with longer distances. When we did this (data not shown) our model performed slightly worse. This indicates that attenuation plays a role in our ability to estimate gaze, and would favor the use of high frequency transducers.

For this proof of concept we chose to average ten individual tests prior to filtering the signal and extracting peak and amplitude. This reduces the eye tracking acquisition speed from a maximum of 2 kHz to 200 Hz, which may not be sufficient to track saccadic eye motion. While this study focused on primarily testing the hypothesis that ultrasound signals can be leveraged to estimate gaze, in future works we will explore avenues to investigate the use for ultrasound in tracking fast eye motion. Specifically, we plan on using a fast-moving model eye coupled with multiple receivers operating at 2 kHz. The GBRT models will be adapted so we can test the potential of ultrasound for fast eye tracking to resolve saccades.

We are interested in investigating the feasibility for using ultrasound sensors for eye tracking in virtual and augmented reality devices. In addition to sampling speed, power consumption is an important factor to consider. The transducers are very low power, in the milliwatt range. Our current system utilizes a high speed A/D converter. This can be replaced with a low power peak detection circuit. On the compute side, GBRTs are considered low compute. This is in particular true for run time on multi-core machines. Specifically, the run time compute complexity for GBRT models is $O(pn_{\mathrm{trees}}/C)$, where *p* represent the number of input features and $n_{\mathrm{trees}}$ are the number of regression trees and *C* is the number of compute cores on a given machine. For $n_{\mathrm{trees}}/C∼1$, the run time complexity for GBRT is on parity with linear regression models, at O(p).

In summary, this study presents data driven proof-of-principle findings to support the claim that ultrasound sensors can be used for gaze estimation.

## Figures and Tables

**Figure 1 sensors-21-04502-f001:**
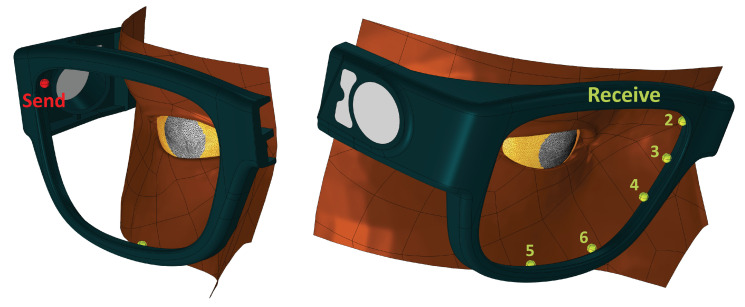
Physical layout for our model. Send location: transducer operating in transmit mode; numbered locations: transducers operating in receive mode. The distance traveled by a ray starting at the send position, reflecting off the cornea, and arriving at receiver 4 is in the 5.28 to 5.35 mm depending on gaze.

**Figure 2 sensors-21-04502-f002:**
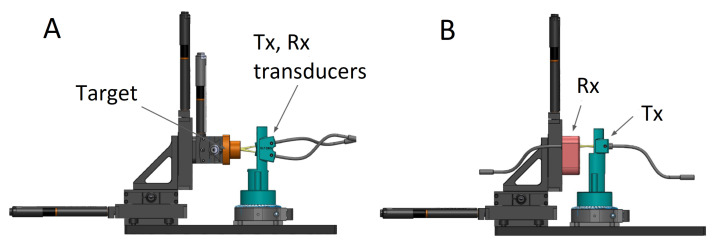
CAD schema for attenuation (**A**) and directionality (**B**) test benches. Tx refers to transducer in transmit mode, Rx receive mode. In A the transducers are fixed and the target is moved. In B the Rx is fixed and the Tx rotated.

**Figure 3 sensors-21-04502-f003:**
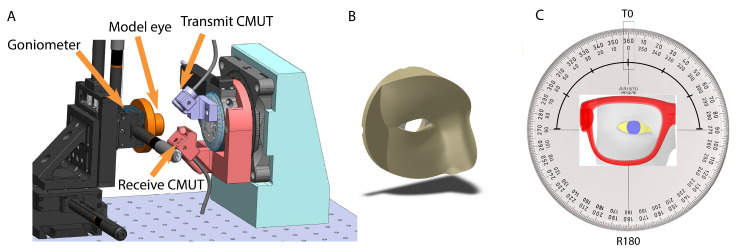
(**A**): CAD schema for the experimental bench-top setup and (**B**): occlusions (**C**). Transducer rotation. The receiver is fixed and the transmitter rotates around an arc. 30 degree steps are shown. We acquired data from −90 to +90 degrees in 10 degree steps.

**Figure 4 sensors-21-04502-f004:**
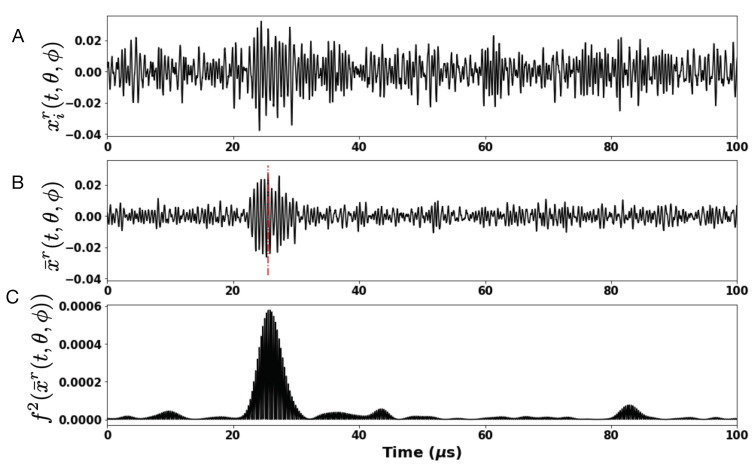
Example of recorded raw time trace of ultrasound sensor signal. The top row (**A**) shows an example of time trace recorded at the receive Ultrasound CMUT sensor in response to a single burst of test signal. The middle row (**B**) shows averaged signal computed from the response to a set of 10 bursts of test signal. Finally the last row (**C**) shows the squared filtered response signal out of a Butterworth filter. The red line indicates the time period of time-to-peak signal detection.

**Figure 5 sensors-21-04502-f005:**
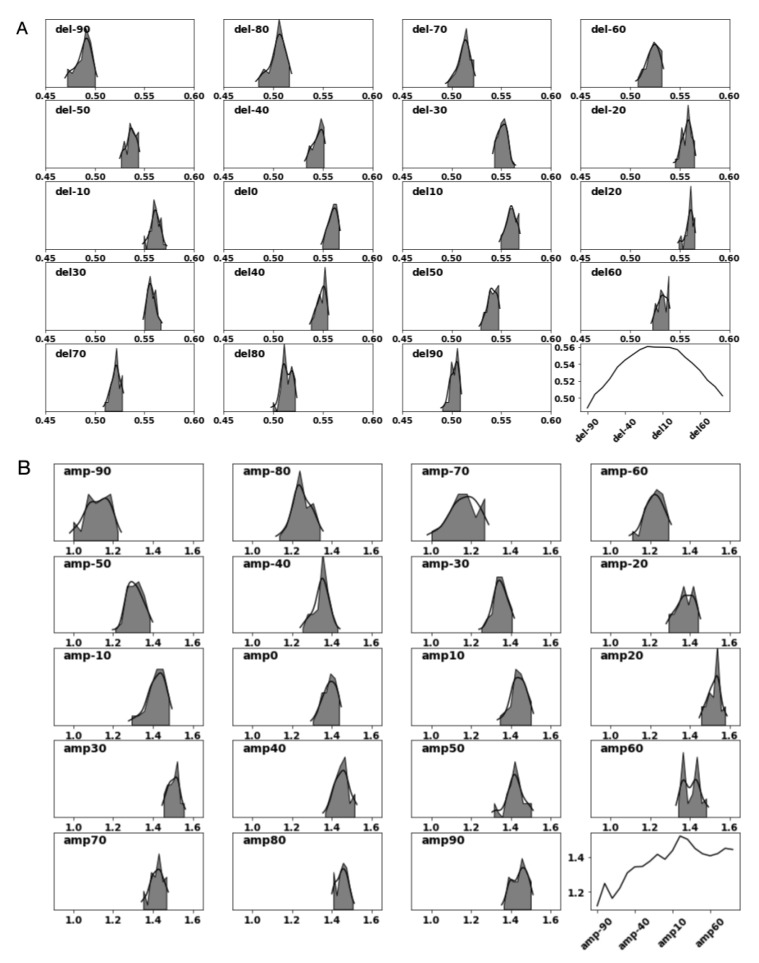
Distribution of the ultrasound time-of-flight (**A**, del/τ) and amplitude (**B**, amp/*a*) signal when the model eye is oriented to gaze angle θ=ϕ=0 degrees.

**Figure 6 sensors-21-04502-f006:**
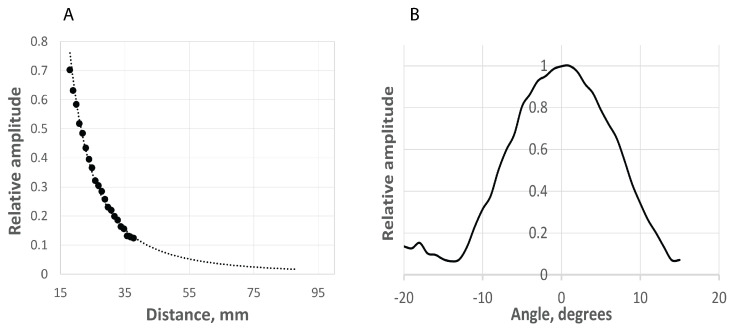
CMUT sensor characterization. (**A**): Attenuation, obtained with Test bench 1, see Figure 2A; (**B**): Directionality, obtained with Test bench 2, see Figure 2B.

**Figure 7 sensors-21-04502-f007:**
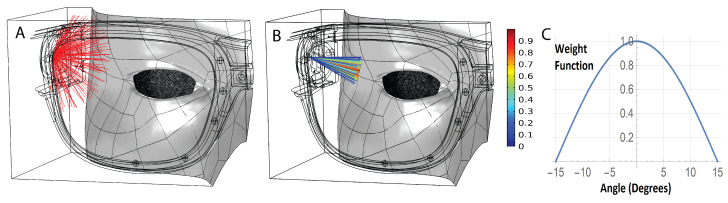
Sensor directionality options. (**A**) Omnidirectional; (**B**) Directional. The color of the ray indicates its intensity; (**C**) Weigh function to mimic the transducer native curve (shown in Figure 6B).

**Figure 8 sensors-21-04502-f008:**
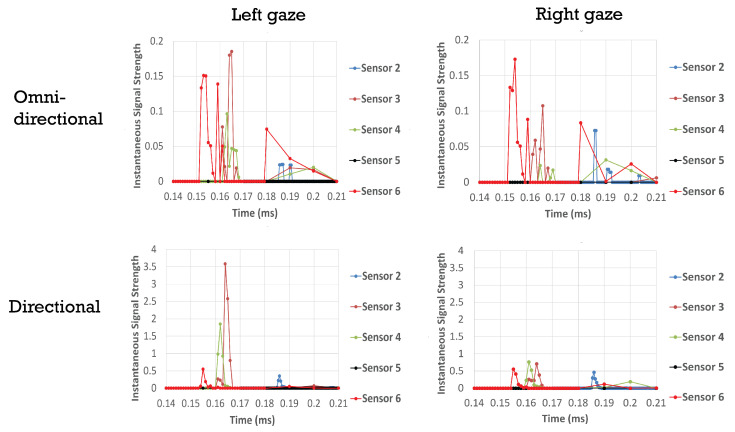
Modeled responses of ultrasound signal measured by the receiver with omni-directional and directional transducers. Left gaze reperesents a 30 degree rotation away from the nose, Rights gaze is 30 degrees towards the nose. Transducer 1 is not shown, as it operates only in transmit mode, see Figure 1.

**Figure 9 sensors-21-04502-f009:**
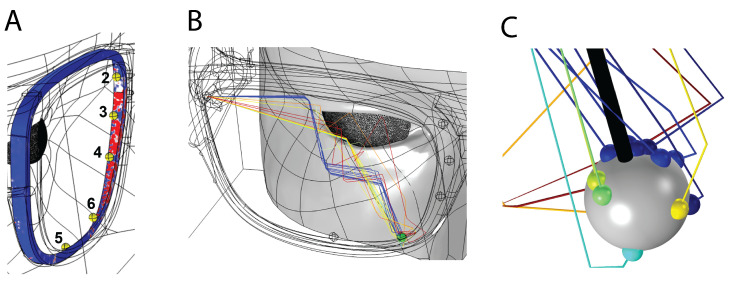
Signal detection: (**A**). Rays reflected from the eye; (**B**). Subset of rays arriving at sensor 6; (**C**). closeup of rays arriving at sensor 6.

**Figure 10 sensors-21-04502-f010:**
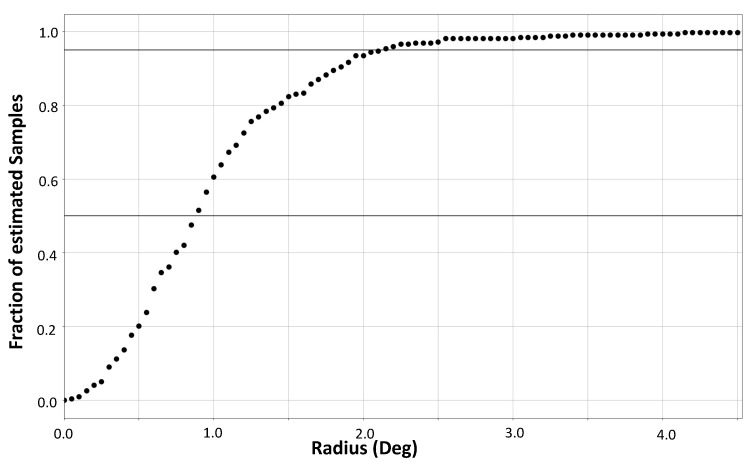
Sensitivity for gaze resolution: fraction of gaze estimates falling within a given radius of corresponding ground truth gaze values.

**Figure 11 sensors-21-04502-f011:**
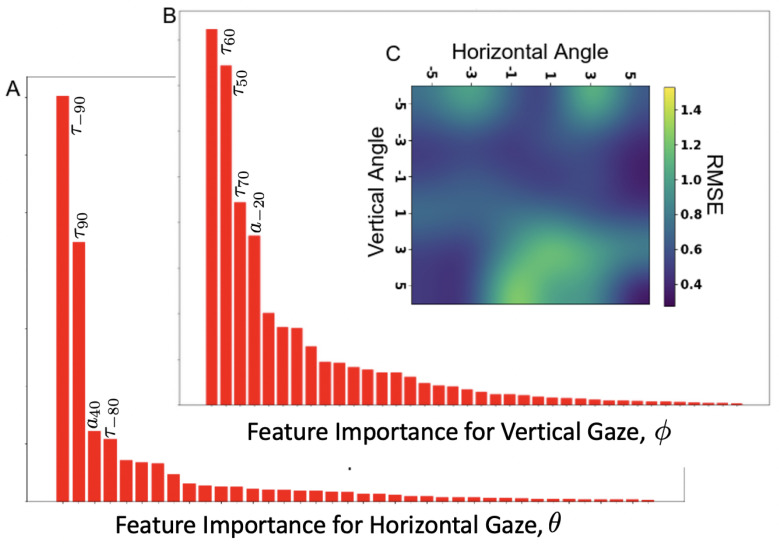
Feature importance and mean accuracy of GBRT models to estimate gaze. (**A**). Horizontal gaze, (**B**). Vertical gaze, (**C**). Error as a function of gaze angle.

**Table 1 sensors-21-04502-t001:** Regression Model for Gaze Estimation. Numbers are presented in terms of mean ± std. dev.

	Adjusted-$R^{2}$	RMSE
Gradient Boosted Tree	90.2 ± 4.6	0.965 ± 0.178
Linear Regression	85.3 ± 7.6	1.177 ± 0.236

**Table 2 sensors-21-04502-t002:** Hyper-parameters for the trained GBRT model. See xgboost parameters in sci-kit learn (https://xgboost.readthedocs.io/en/latest/parameter.html accessed on 5 April 2021) for explanation of these hyper-parameters.

Hyper-Parameters (XGBoost GBRT Model)	
learning rate	0.0825
max tree depth	5
# regression trees	750
min. child weight	23
α regularization	0.01
λ regularization	1

## Data Availability

None.

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
