# Peer review of "Ultrasound for Gaze Estimation—A Modeling and Empirical Study†"

_sensors, 2021, doi:10.3390/s21134502_

Round 1
Reviewer 1 Report
*Summary*
The paper empirically studies the effectiveness for gaze estimation in augmented reality (AR) and virtual reality (VR) applications using ultrasound transducers. Experiments demonstrate that these transducers can effectively produce signals useful to resolve eye gaze.
*Major issues*
Section 2 would benefit from more explanation as to the choices of model (GBRT) and performance metrics (R^2 and RMSE). Ideally, GBRT would be benchmarked against a simple baseline estimator (e.g., a single regression tree or linear regression) to demonstrate a performance benefit from using a boosting method. A comparison of the computational demand of GBRT versus a simple baseline estimator would also be useful, especially because it is stated “GBRTs are considered low compute.”
*Minor issues/suggestions*
Pg. 1, line 18: “third Purkinje” → “third Purkinje images”
Pg. 2, line 41: “Occlusions” → “Eye occlusions”?
Pg. 5, line 173: why is it that two separate GBRT models are trained to estimate the response? Is it because there are two responses (horizontal and vertical gaze)?
Pg. 9, Table 1: table notes should define some of the non-self explanatory hyper-parameters; e.g., what is min. child weight and alpha/lambda regularization?
Author Response
The paper empirically studies the effectiveness for gaze estimation in augmented reality (AR) and virtual reality (VR) applications using ultrasound transducers. Experiments demonstrate that these transducers can effectively produce signals useful to resolve eye gaze.
*Major issues*
Section 2 would benefit from more explanation as to the choices of model (GBRT) and performance metrics (R^2 and RMSE). Ideally, GBRT would be benchmarked against a simple baseline estimator (e.g., a single regression tree or linear regression) to demonstrate a performance benefit from using a boosting method. A comparison of the computational demand of GBRT versus a simple baseline estimator would also be useful, especially because it is stated “GBRTs are considered low compute.”
We have revised our manuscript to also present findings on gaze estimation model trained using a linear regressor. The findings are reported in Table 2 (see lines 284-290). Section 2.3.2 is revised to comment on our rationale for choice of GBRT as a regression model. We also comment on the choice of RMSE and R2 measures for model evaluation (lines 194-203). Finally, discussion section is revised to further elaborate on the low compute aspect of GBRT models (lines 346-351).
*Minor issues/suggestions*
Pg. 1, line 18: “third Purkinje” → “third Purkinje images”
Edited.
Pg. 2, line 41: “Occlusions” → “Eye occlusions”?
Edited.
Pg. 5, line 173: why is it that two separate GBRT models are trained to estimate the response? Is it because there are two responses (horizontal and vertical gaze)?
Edited.
Pg. 9, Table 1: table notes should define some of the non-self explanatory hyper-parameters; e.g., what is min. child weight and alpha/lambda regularization?
We reference relevant literature on gradient boosted tree implementation in python package sci-kit learn that define the stated hyper parameters. Specifically, min-child weight corresponds to the minimum number of instances (samples) needed to be in the leaf-node before it is split. Alpha corresponds to the L1 regularization term on weights and Lambda corresponds to the L2 regularization term on weights.
Reviewer 2 Report
Authors proposed interesting approach for AR devices with the eye-gaze function using ultrasound devices. The data with different angle with face modeling is good. The measured amplitude with different angles looks reasonable for AR devices. The signal detection parameters with sensitivity after 2.0 degrees looks good. Overall, the manuscript is well written. However, there are some missing references. Therefore, the manuscript could be minor revision if authors follow the suggestions.
1. Please correct "40kHz" to "40 kHz".
2. Please correct "500kHz-2MHz" to "500 kHz - 2 MHz".Please check others.
3. Please correct "CAD schema" to "CAD schem a".
4. In Figure 4c, I am wondering the x-axis has frequency domain. It looks like spectrum data after Butteworth filter.
5. What is y-axis unit for Figure 6B ?
6. In Line 248, figure 4 -> Figure 4.
7. Please provide the reference for the sentence (This type of transducer has found numerous medical applications in both imaging and therapy, which are applications for contact ultrasound) with the reference (Khuri-Yakub, Butrus T., and Ömer Oralkan. "Capacitive micromachined ultrasonic transducers for medical imaging and therapy." Journal of micromechanics and microengineering 21.5 (2011): 054004. ) or another reference.
8. Please provide the reference for the sentence (In this mode, the difference in impedance between air and tissue means over 99 percent of the ultrasound signal will be reflected by the eye surface) with the reference (Kim, K., & Choi, H. (2021). High-efficiency high-voltage class F amplifier for high-frequency wireless ultrasound systems. PloS one, 16(3), e0249034. ).
9. Please provide the reference for the sentence (To estimate the reflection of the signal we used the formula) with the reference (Zhou, Qifa, et al. "Piezoelectric single crystal ultrasonic transducers for biomedical applications." Progress in materials science 66 (2014): 87-111. ) or another reference.
10. In the reference section, authors should use proper reduced format for journal papers.
11. What are the unit of y-axis in Figure 5 ?
Author Response
Authors proposed interesting approach for AR devices with the eye-gaze function using ultrasound devices. The data with different angle with face modeling is good. The measured amplitude with different angles looks reasonable for AR devices. The signal detection parameters with sensitivity after 2.0 degrees looks good. Overall, the manuscript is well written. However, there are some missing references. Therefore, the manuscript could be minor revision if authors follow the suggestions.
- Please correct "40kHz" to "40 kHz".
Corrected.
- Please correct "500kHz-2MHz" to "500 kHz - 2MHz".Pleasecheck others.
Corrected.
3. Please correct "CAD schema" to "CAD schema".
Our spelling is correct.
- In Figure 4c, I am wondering the x-axis has frequency domain. It looks like spectrum data afterButteworthfilter.
No, this is time domain.
5. What is y-axis unit for Figure 6B ?
Figure replaced, y-axis edited.
- In Line 248, figure 4 -> Figure 4.
Corrected.
- Please provide the reference for the sentence (This type of transducer has found numerous medical applications in both imaging and therapy, which are applications for contact ultrasound) with the reference (Khuri-Yakub, Butrus T., andÖmerOralkan. "Capacitive micromachined ultrasonic transducers for medical imaging and therapy." Journal of micromechanics and microengineering 21.5 (2011): 054004. ) or another reference.
Suggested reference added.
8. Please provide the reference for the sentence (In this mode, the difference in impedance between air and tissue means over 99 percent of the ultrasound signal will be reflected by the eye surface) with the reference (Kim, K., & Choi, H. (2021). High-efficiency high-voltage class F amplifier for high-frequency wireless ultrasound systems. PloS one, 16(3), e0249034. ).
The amount reflected is obtained with the formula mentioned in the next comment. We added a reference for it. The suggested citation is focused on developing electronics, not methods for estimating reflections.
- Please provide the reference for the sentence (To estimate the reflection of the signal we used the formula) with the reference (Zhou, Qifa, et al. "Piezoelectric single crystal ultrasonic transducers for biomedical applications." Progress in materials science 66 (2014): 87-111. )or another reference.
We added a citation already part of this manuscript. It deals with reflection from the cornea.
- In the reference section, authors should use proper reduced format for journal papers.
Edited.
- What are the unit of y-axis in Figure5 ?
Figure 5 represents the distribution plot for the ultrasound time-of-flight and the amplitude signal at the receiver. As such, the y-axis refers to the count of for samples falling within a given epsilon bin for these signals and therefore y-axis does not have any units.
Reviewer 3 Report
Overall, I think this is a promising piece of work. Existing mobile eye trackers tend to be expensive (except maybe the Pupil Labs system) and perform poorly in changing lighting conditions (e.g., going from indoors to outdoors). While dispersion algorithms can split raw eye tracking signals fairly accurately into saccades and fixations at lower sampling frequencies, they do not allow for measurements of properties of saccadic eye movements (e.g., any deviations in the path due to attention shifts).
I have two main concerns with the paper in its current form: (1) it is very technical, and may therefore be difficult to follow for those interested in eye tracking, but not being experts in sensor-technology (like myself). Below I indicate what sections could be difficult to follow for this group of readers. I think some additional clarifications could go a long way here, (2) the technology has not yet been tested on human eyes, and it is unclear whether it can be expected that the system will indeed outperform currently available eye trackers such as the Tobii 2 glasses or the Pupil labs system. While I can see that it may be beyond the scope of the current contribution to perform such comparisons with existing systems, I think some further details are needed about what can be expected and how the results compare to what can be found for existing systems. Some of my line-by-line comments below may help finding where in the paper such information could be added.
Line-by-line comments:
Lines 15-30. This is a clear and well-argued introduction about why infrared eye tracking is needed, and what limitations of traditional light-based eye tracking it may overcome.
Line 33. It is unclear what flight data mean in this context: Is this linked to the trajectory of eye movements?
Line 38. It is unclear what is mean with ‘user-specific placement’ of sensors in the context of AR and VR. Are the glasses adjusted to each individual observer?
Line 40. It is unclear what kind of occlusion is spoken about. Is this the closure of the eye-lid during blinks? Does this mean the new device can measure REM-sleep related eye movements?
Line 42. More explanation of what the 40kHZ transducers are may be needed for those interested in measuring eye movements, but less skilled in engineering. The same holds for ‘finite-element-modeling' in line58 and ‘directional and omnidirectional transmit’ in line 60.
Line 60. Wouldn’t it be important to also test the technology on human eyes, comparing the accuracy with already existing light-based eye trackers?
Line 69. Here it may be important to relate to what eye tracking specialists may already know: calibration of the eye tracker (using regression models). Best add a few lines to explain why the machine learning is needed here, as it may not be clear to those primarily interested in eye tracking.
Line 78. Here I would first explain the general principle of the system before going into the details.
Line 128. Do your occlusions mean cast shadows?
Line 131. In terms of human eye movements, can you explain what kind of movements you tested with your model? Were the tested movements as fast as saccades? What directions of movements did you test?
Line 136. Please explain the general reason for engineering features here (why couldn’t you use the raw signal?). What considerations were made when constructing features from the raw sensor signal?
Line 166. Can you explain why you choose boosted trees over other available regression models?
Figure 5. Are these static eye positions? How does the signal of the sensors relate to the direction and speed of the eye?
Line 179. Is this the standard loss function used for boosted trees? How do you work around possible issues with local minima of the loss function?
Line 186. Can you explain why the distance-signal function is important for eye tracking? Shouldn’t the glasses in which the sensors are integrated be at a relatively constant distance? The role of incident angle for eye tracking is easier to grasp.
Line 200. Please explain how the tested movements relate to eye movements, such as saccades or smooth pursuit.
Figure 8. In the caption: Can you add why there is no senser 1? Can you add what is meant with left gaze and right gaze (different eyes? Different direction?). The signal seems to be zero most of the time; can you add why this may be? When it is not zero, it seems to be rather noisy. Can you add what this may reflect?
Line 212. Can you explain what the weight function is used for? Why is a weight function needed? Is the signal in Figure 8 often zero due to the weight function? Does the same hold for the noisy signal?
Line 256. For the boosted tree model, do you also use a validation / test set? How was the original set split into training and test? Were strata used? (e.g., for gaze direction). Was one split in training and test sufficient for this set size?
Line 262. Shouldn’t the error bias be symmetric? If the eye tracker would be used for looking at a computer screen, what sections of the screen would have poorer recording of eye movements? Would you be able to compensate by combining eye movements recorded in both eyes?
Figure 11. How to interpret the feature importance graphs? Are the results according to what could be expected?
Line 318. Can you add something about the expected cost of the system? How would it compare to standard eye trackers? Are you planning to conduct studies with actual eye movements and compare the accuracy with those in commercially available eye trackers, such as the Tobii 2 glasses or the Pupil Labs system?
Author Response
Overall, I think this is a promising piece of work. Existing mobile eye trackers tend to be expensive (except maybe the Pupil Labs system) and perform poorly in changing lighting conditions (e.g., going from indoors to outdoors). While dispersion algorithms can split raw eye tracking signals fairly accurately into saccades and fixations at lower sampling frequencies, they do not allow for measurements of properties of saccadic eye movements (e.g., any deviations in the path due to attention shifts).
I have two main concerns with the paper in its current form: (1) it is very technical, and may therefore be difficult to follow for those interested in eye tracking, but not being experts in sensor-technology (like myself). Below I indicate what sections could be difficult to follow for this group of readers. I think some additional clarifications could go a long way here, (2) the technology has not yet been tested on human eyes, and it is unclear whether it can be expected that the system will indeed outperform currently available eye trackers such as the Tobii 2 glasses or the Pupil labs system. While I can see that it may be beyond the scope of the current contribution to perform such comparisons with existing systems, I think some further details are needed about what can be expected and how the results compare to what can be found for existing systems. Some of my line-by-line comments below may help finding where in the paper such information could be added.
This paper provides the first proof of principle for using ultrasound to estimate gaze. This was done with a static model eye. From this we want to move to dynamic recordings with a moving model eye, followed by tests on human subjects. At this stage our system cannot be compared to commercial eye trackers in terms of performance. We believe it has potential to measure saccades in that it can operate at 2 kHz. We only measure two parameters per sensor, so computing a gaze estimation is not compute-intensive.
We have added introductions to some of the concepts and why we did what we did. We hope these will make the writing more approachable.
Line-by-line comments:
Lines 15-30. This is a clear and well-argued introduction about why infrared eye tracking is needed, and what limitations of traditional light-based eye tracking it may overcome.
Line 33. It is unclear what flight data mean in this context: Is this linked to the trajectory of eye movements?
We added text to explain this later in the paragraph. “Distances are measured with each transducer based on the time for it to receive the reflection of its own signal. Given the cornea protrudes, this time changes with the gaze angle.”
Line 38. It is unclear what is mean with ‘user-specific placement’ of sensors in the context of AR and VR. Are the glasses adjusted to each individual observer?
See added text. “In AR/VR the eye tracking system will be fixed and the position of the eye will vary from user to user, which means alignment will vary.”
Line 40. It is unclear what kind of occlusion is spoken about. Is this the closure of the eye-lid during blinks? Does this mean the new device can measure REM-sleep related eye movements?
We added text to explain our occlusions, which consist of static eyelids. Our goal is to measure gaze (when the eyes are open). Using it to detect REM is an interesting idea. Given that our signals are reflected from the first surface they encounter, and REM results in deformation of the eyelids, this may be doable.
Line 42. More explanation of what the 40kHZ transducers are may be needed for those interested in measuring eye movements, but less skilled in engineering. The same holds for ‘finite-element-modeling' in line58 and ‘directional and omnidirectional transmit’ in line 60.
An explanation of the 40 kHz transducers has been added. The typical sensing range for 40-70kHz transducers would be in meters with a resolution of 1 cm, which would be insufficient to estimate gaze. We also added text to explain why we chose the type of simulation we did, see Line 78. Omnidirectional and directional have their common meaning.
Line 60. Wouldn’t it be important to also test the technology on human eyes, comparing the accuracy with already existing light-based eye trackers?
As explained in the general comments, this will need to be done. We first wanted to determine if we could estimate static gaze using a model eye. Dynamic gaze estimation using a moving model eye is next, followed my human studies.
Line 69. Here it may be important to relate to what eye tracking specialists may already know: calibration of the eye tracker (using regression models). Best add a few lines to explain why the machine learning is needed here, as it may not be clear to those primarily interested in eye tracking.
We have edited the section in question to comment on the requirements for training a nonlinear machine learning model for gaze estimation. See lines 73-80
Line 78. Here I would first explain the general principle of the system before going into the details.
We added text. Ultrasound is modeled as rays released all at once from a single point. Their position is updated at fixed time intervals. We did this so we could trace the path of signals reaching the receiver and determine if they were reflections from the cornea or the skin or glasses.
Line 128. Do your occlusions mean cast shadows?
W added an explanation. “This is a step forward from previous modeling which totally ignored occlusions. We did not model or attempt to integrate eyelashes.”
Line 131. In terms of human eye movements, can you explain what kind of movements you tested with your model? Were the tested movements as fast as saccades? What directions of movements did you test?
All our tests were on static model eye positions.
Line 136. Please explain the general reason for engineering features here (why couldn’t you use the raw signal?). What considerations were made when constructing features from the raw sensor signal?
We added an explanation about why this is needed. “The raw signal carries too much noise to allow for accurate peak time and amplitude measurements. Improvements are possible (data not shown). For this proof of concept we used averaging and filtering.”
Line 166. Can you explain why you choose boosted trees over other available regression models?
As noted in our response to reviewer 1 comments, gradient boosted trees are low compute nonlinear models most suited for regression tasks where the input features are available. The simplest model is linear regression and as we show in Table 2, the nonlinearity imposed by occlusions necessitate the use of nonlinear regression model to get the best performing gaze estimation model for the dataset under consideration.
Figure 5. Are these static eye positions? How does the signal of the sensors relate to the direction and speed of the eye?
These are for static position. Variability reflects system noise.
Line 179. Is this the standard loss function used for boosted trees? How do you work around possible issues with local minima of the loss function?
Minimum squared error loss (also called L2 loss) is the standard choice for regression problem such as gaze estimation in general. We choose to use the same loss function for GBRT model training as well. Hyper-parameter optimization is one possible work around for issues with local minima and we adopted a grid search hyper-parameter selection strategy to select the best hyper-parameters for training the GBRT model.
Line 186. Can you explain why the distance-signal function is important for eye tracking? Shouldn’t the glasses in which the sensors are integrated be at a relatively constant distance? The role of incident angle for eye tracking is easier to grasp.
See expanded second paragraph in the introduction.
Line 200. Please explain how the tested movements relate to eye movements, such as saccades or smooth pursuit.
As stated before we did not measure movements. We believe this technology has the potential to resolve saccades as it can operate at up to 2 kHz.
Figure 8. In the caption: Can you add why there is no senser 1? Can you add what is meant with left gaze and right gaze (different eyes? Different direction?). The signal seems to be zero most of the time; can you add why this may be? When it is not zero, it seems to be rather noisy. Can you add what this may reflect?
Added an explanation for numbering. Transducer number 1 is used as a source only, see Figure 1. No signal means there are no rays reaching the receiver. When we do see a signal it is the sum of all signals detected. All take different paths to reach the receiver and so will not arrive at the same time.
Line 212. Can you explain what the weight function is used for? Why is a weight function needed? Is the signal in Figure 8 often zero due to the weight function? Does the same hold for the noisy signal?
The signal strength of our transducers varies with the angle to the normal. We determined this experimentally in Figure 6B. We mimic the measured curve with a function as shown in Figure 7. This is to match our model to experimental data.
Line 256. For the boosted tree model, do you also use a validation / test set? How was the original set split into training and test? Were strata used? (e.g., for gaze direction). Was one split in training and test sufficient for this set size?
As reported in the paper, we collect a total of 1620 samples for training GBRT model. We choose 5-fold cross-validation methodology to train and validate GBRT model. We report our findings in terms of mean +/- std. Dev for the performance metrics computed over the 5-folds.
5-cross validation strategy works as follows: We divide the entire data into 5 equal buckets. For each fold training, we select the 4 of 5 subsets to train the model and the 5th subset as a hold out dataset for validation. For subsequent fold model training we iterate of the remainder of the 4 subsets. Performance is reported as average (and std. dev) across all 5 cross-validation datasets.
Line 262. Shouldn’t the error bias be symmetric? If the eye tracker would be used for looking at a computer screen, what sections of the screen would have poorer recording of eye movements? Would you be able to compensate by combining eye movements recorded in both eyes?
I do not see how the comment relates to the line. The error would be expected to be symmetric in the absence of occlusions. These are asymmetrical and affect our measurements. Using bilateral information should help once we have such a full eye tracker built based on this technology.
Figure 11. How to interpret the feature importance graphs? Are the results according to what could be expected?
As the name suggests, the feature importance graph plots the features that the model deems most important to explain the data (in this case, minimize the L2 loss on ground truth gaze vectors). The purpose of this graph is to better understand of all the input features, which features (as represented by the height of the bar) are most important to explain the data. A simple way to think of features is in terms of the principal components in PCA analysis.
While we expect the time-of-flight signal to be most relevant to the current application, it is apriori difficult to gauge which sensor time-of-flight data is most relevant for gaze estimation.
As the chart shows, for horizontal gaze, the sensors at location 50 and 60 degrees contribute most signal whereas for vertical gaze, sensors at extreme locations –90 and 90 degrees contribute the most signal. These results are difficult to apriori anticipate and may be the result of as yet not well understood nonlinear interactions of reflected ultrasound waves captured by the various sensors.
Line 318. Can you add something about the expected cost of the system? How would it compare to standard eye trackers? Are you planning to conduct studies with actual eye movements and compare the accuracy with those in commercially available eye trackers, such as the Tobii 2 glasses or the Pupil Labs system?
It is too early for us to speculate on the cost of an eye tracking system based on ultrasound. When produced at scale, the sensors are inexpensive. The same sensors are embedded in the disposable tip of the Otonexus ultrasound otoscope (https://otonexus.com/). However, the sensors are only part of the cost of the system build.
Round 2
Reviewer 1 Report
The authors have sufficiently addressed my comments.